# Inner-Shell Photodetachment of Na⁻ Using R-Matrix Methods

**T. W. Gorczyca [1],\*** , **H.-L. Zhou [2]**, **A. Hibbert [3]**, **M. F. Hasoglu [4]** and **S. T. Manson [2]**

[1] Department of Physics, Western Michigan University, Kalamazoo, MI 49008, USA
[2] Department of Physics and Astronomy, Georgia State University, Atlanta, GS 30303, USA; phyhlz@gmail.com (H.-L.Z.); smanson@gsu.edu (S.T.M.)
[3] School of Mathematics and Physics, Queen's University of Belfast, Belfast BT7 1NN, UK; a.hibbert@qub.ac.uk
[4] Department of Computer Engineering, Hasan Kalyoncu University, Sahinbey 27100, Gaziantep, Turkey; mfatih.hasoglu@hku.edu.tr
\* Correspondence: thomas.gorczyca@wmich.edu

**Abstract:** Inner-shell photodetachment of Na⁻ near the L-edge threshold was investigated using the R-matrix method. Significant structure was found in the cross section, and this structure is shown to be related to the complicated correlated electron dynamics endemic in negative ions. Comparison with experiment suggests that the absolute values of the measured cross section might be too small by a factor of two.

**Keywords:** photodetachment; inner-shell phenomena

## 1. Introduction

Photodetachment is a highly correlated process [1–5]. Correlation is required to accurately characterize the initial state wave function of a negative ion, or even to get the initial state bound in many cases. As a well-studied example, the (uncorrelated) Hartree-Fock energy level for the simplest H⁻ anion lies above the hydrogen atomic energy level [6]. For the final state wave function in the photodetachment process, correlation in the form of interchannel coupling has been found to be quite important. Since correlation dominates the photodetachment process, studying photodetachment is an excellent venue for understanding the correlated dynamics of electrons in atomic negative ions. Studies of outer-shell photodetachment have a long history [1–5]. More recently, principally over the past two decades, investigations of inner-shell photodetachment have been performed, both theoretical and experimental, particularly He⁻ and Li⁻ [7–16]. Much has been done on heavier anions, as well; see, e.g., Refs. [17,18] and references therein. Inner-shell photodetachment of atomic anions is also of particular interest because the initial-state wave function of the transition remains almost exactly the same as in the neutral atom, but the final-state wave functions can differ dramatically. This is due to rather different fields experienced in the anion and neutral atom by the emerging photoelectron. The differences of the inner-shell cross section between the atom and the negative ion are entirely due to the different outer-shell electronic structure, which is highly correlated in the case of a negative ion. This correlated photodetachment process can be used to quantify the spectroscopy of the tenuously bound valence electrons, a photoelectron probe of the many-body dynamics inherent in the binding of the negative ion. The simplest multielectron negative ions, beyond the tractable two-electron H⁻ case, are He⁻ and Li⁻ and their photodetachment has been dealt with extensively already [7–16].

The inner-shell photodetachment of Na$^-$, which lies below Li$^-$ in the periodic table, has been studied experimentally [19], and calculations that included many-body dynamics have been performed [20,21]. Neither of those calculations could reproduce, even qualitatively, the sharp resonance that is attributed to a multiple-electron excitation process [19]. This means that important physics was not included in those calculations. To improve upon the situation, R-matrix methods are used here for the calculations since they are able to account for initial state correlations and final state interchannel coupling, thereby including coupling with detachment-plus-excitation channels. A version of the Belfast R-matrix methodology that has been modified to accommodate photodetachment has been employed [15].

In the next section, a brief discussion of the theoretical methods employed is given, along with the results of our calculations of the relevant discrete state energies of Na and Na$^-$. The following section presents the photodetachment cross section and a comparison with experiment and previous calculations. The final section presents a summary and conclusions.

## 2. Theoretical Methods

In carrying out the present R-matrix calculations for the Na$^-$ photodetachment near the $2p$ thresholds, we begin with the exact same structure as used in our earlier R-matrix study for Na$^-$ [22]. To recap briefly, an atomic orbital basis is generated using the atomic structure program CIV3 [23]. The $1s$, $2s$, $2p$, and $3s$ orbitals are obtained from a Hartree-Fock calculation for the $1s^2 2s^2 2p^6 3s$ ground state of the Na target, whereas the $(3p, 3d, 4s, 4p, 4d,$ and $4f)$ orbitals are each optimized on their corresponding frozen-core $1s^2 2s^2 2p^6 nl$ states ($n = 3, 4$), completing the physical, outer-shell-optimized orbital basis for the lowest Na target states. An additional basis of pseudoorbitals ($5s, 5p, 5d,$ and $5f$) is obtained from multi-configuration optimizations on the $1s^2 2s^2 2p^5 3l 3l'$ inner-shell vacancy states, thus adding consistent correlation additions to both the Na and Na$^-$ states.

The 14 CIV3 orbitals, 10 physical and 4 treated as correlation pseudoorbitals, are used to construct the first 7 singly-excited Na target states, and 37 inner-shell-excited states. These are all bounded within the R-matrix radius determined to be 56.2 a.u. A separate, orthogonal basis of 50 continuum orbitals is used to represent the photodetached electron beyond the R-matrix radius. The "bound" and "continuum" orbitals are then coupled together to represent all Na$^-$ states: the initial $1s^2 2s^2 2p^6 3s^2 (^1S)$ Na$^-$ state, the e$^-$ + Na scattering states, and the inner-shell-excited resonance states such as the dominant resonance just below the $1s^2 2s^2 2p^5 3s4s$ threshold, as will be seen.

We also include an additional, important component in the calculations, that of a pseudoresonance removal procedure [24]. A highly-correlated R-matrix basis, without proper care, can lead to an over-completeness of the bound $N + 1 = 12$-electron Na$^-$ bound or quasibound (resonance) state basis, or the e$^-$ + Na scattering states. This in turn causes unphysical *pseudoresonances* to arise in the computed cross sections. The R-matrix method usually uses an orthonormal orbital basis, constraining the continuum orbital basis to be orthogonal to the bound-orbital basis. To compensate for the subspace projected out of the wavefunction Hilbert space by this enforced orthogonality, additional, so-called $(N + 1)$-electron configurations must be added back into the full wavefunction basis. However, the exact choice of linear combinations of $(N + 1)$-electron configurations needed requires examination. A linear algebra method for choosing the minimum basis needed to span the $N + 1 = 12$-electron states determines the minimal rotated basis needed, yielding a reduced basis of linear combinations of configurations. The "continuum" orbitals are generated in the first step of the R-matrix calculation [25,26] by using a model core potential and appropriate surface boundary conditions, along with necessary Laguerre or Gram-Schmidt orthogonalization to the bound-orbital basis.

The resultant energies for the Na 11-electron target states are shown in Table 1, and compared to the NIST values [27]. We only use doublet states of Na, not quartet states or higher, since our final-symmetry $^1P^o$ singlet states must be composed from a doublet electron coupled to the Na target state. In addition, shown in Table 1 is the electron affinity of Na (photodetachment threshold energy)

along with experiment [28]. From this table, it is evident that the agreement between theory and experiment for both the electron affinity and the states of the neutral Na atom is quite good.

**Table 1.** R-matrix energies of the Na target states compared to the NIST values [27], the NIST Na$^+$ ground state energy, and the R-matrix electron affinity of Na$^-$ compared to experiment [28].

| Term State | R-matrix (Ryd) | EXP (Ryd) |
|---|---|---|
| Na$^-$ $2p^63s^2$ | $-0.0395$ | $-0.0403$ |
| $2p^63s(^2S)$ | 0.00000 | 0.00000 |
| $2p^63p(^2P)$ | 0.15827 | 0.15462 |
| $2p^64s(^2S)$ | 0.23623 | 0.23456 |
| $2p^63d(^2D)$ | 0.26720 | 0.26584 |
| $2p^64p(^2P)$ | 0.27809 | 0.27585 |
| $2p^65S(^2S)$ | | 0.30255 |
| $2p^64d(^2D)$ | 0.31606 | 0.31483 |
| $2p^64f(^2F)$ | 0.31651 | 0.31518 |
| $\vdots$ | | |
| Na$^+$ $2p^6(^1s)$ | | 0.37772 |
| $\vdots$ | | |
| $2p^53s^2(^2P)$ | 2.26134 | 2.26550 |
| $2p^53s3p(^2D)$ | 2.47510 | 2.46049 |
| $2p^53s3p(^2P)$ | 2.47950 | 2.46751 |
| $2p^53s3p(^2S)$ | 2.50273 | 2.48991 |
| $2p^53s3p(^2S)$ | 2.58706 | 2.55864 |
| $2p^53s3p(^2D)$ | 2.58870 | 2.55624 |
| $\vdots$ | | |

## 3. Results and Discussion

To understand the physics of what goes on in the vicinity of each threshold, it is important to keep in mind that near-threshold negative-ion photodetachment cross sections behave very differently from cross sections for photoionization of neutral systems or positive ions. First of all, in the photoionization process, there are an infinity of autoionizing (Feshbach) resonances below each threshold [29], while in photodetachment there generally are none or, at most, one or two [5]. Secondly, while photoionization cross sections have a finite value at threshold [29], photodetachment cross sections are zero at threshold, which means that photodetachment cross sections always rise from threshold [5]. The results of the photodetachment cross section calculations over a broad L-edge energy region are depicted in Figure 1, along with the experimental results [19]; the qualitative agreement is excellent. All of the structure observed in the experiment is also seen in the calculated results, including the resonance maximum just above the first inner-shell threshold, the structure associated with the $2p^53s3p$ thresholds, the sharp Feshbach resonance just above 36 eV, and the gradual rise of the cross section with energy above these resonances.

The maximum in the cross section just above the first inner-shell threshold, at about 31.3 eV, is the result of the ordinary photodetachment rise from zero, plus a resonance which we assign to be the $2p^53s^2(\epsilon s, ns)$ resonance. This occurs because in the negative ion, the $2p^53s^2ns$ excitation lies *above* the $2p^53s^2$ threshold. This is similar to what was found recently in Li$^-$ [30]. This resonance has both shape and Feshbach characteristics, which demonstrates how different negative ions are compared to their neutral atom counterparts; even the language of neutral atoms is inadequate to describe certain negative ion phenomena. In addition, it is evident that the calculation seriously overestimates the size of this near-threshold resonance. This occurs because the present calculation omits the post collision interaction (PCI) effect [31,32], whereby a slow (near-threshold) photoelectron from an inner shell can be recaptured and the energy transferred to a fast photoelectron associated with photoemission from an outer shell via interchannel coupling. This was found to be quite important in the threshold

behavior of the inner-shell photodetachment of Li$^-$ [14,15]. As discussed below, however, there are reasons to believe that PCI is not very important in this case.

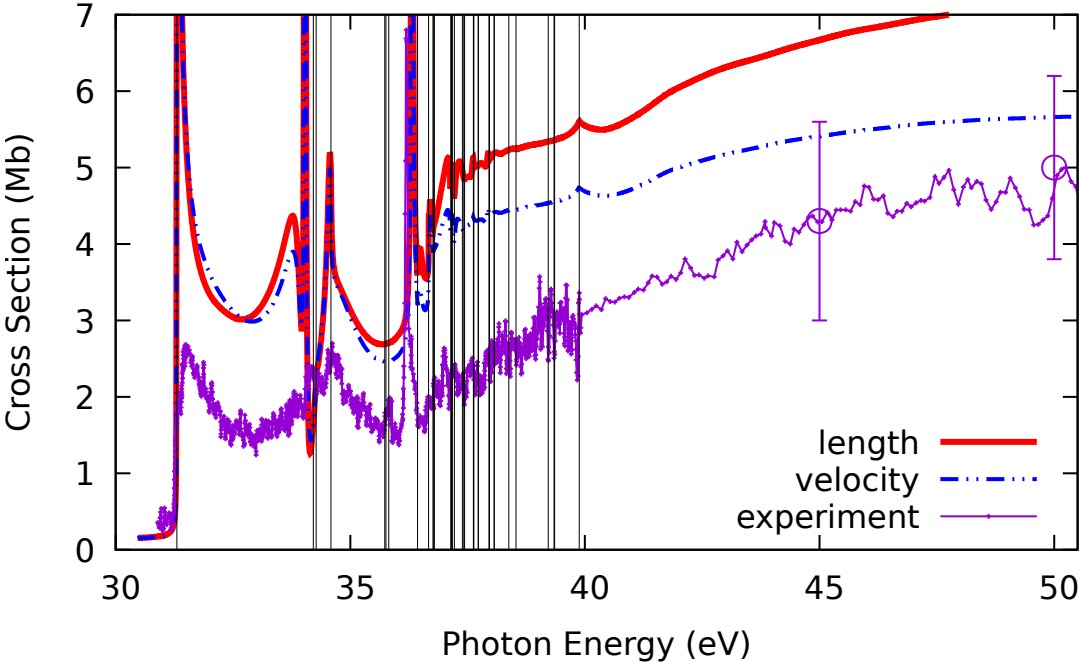

**Figure 1.** Calculated R-matrix Na$^-$ photodetachment cross sections compared to experiment [28]. The calculated inner-shell thresholds are shown as vertical black lines.

In the region of the $2p^53s3p$ thresholds—in the 34 eV region—the calculated results exhibit a complex series of structures that is only hinted at in the somewhat unresolved experimental results. The calculation does, however, reproduce the valleys in the cross section both between the threshold maximum and the $2p^53s3p$ threshold regions, and above this region, as well. It is likely that one of the resonances that theory predicts is too high here due to associated PCI effects. In any case, the fact that length and velocity results are almost identical in this region lends credence to the theoretical predictions, thereby suggesting that this energy region be scrutinized with greater experimental resolution.

Between the sharp resonance just above 36 eV and about 40 eV, theory shows a number of small resonances associated with the many inner-shell thresholds in this region. While it is not entirely clear, due to the finite resolution, the experimental results also seem to show a number of resonances, as well. Above about 40 eV, the cross sections, both theoretical and experimental, are more or less smooth and increasing. The increasing behavior arises from the $2p \rightarrow \epsilon p$ shape resonance that is also known in the inner-shell photoionization of neutral Na [33]. Furthermore, with increasing energy, it is reasonable to expect that the $2p$ photodetachment cross section should approach the $2p$ photoionization cross section of neutral Na. This is because, as mentioned earlier, the initial state $2p$ wave functions are essentially the same in both cases, and in the final states, a fast (as opposed to a slow, near-threshold) photoelectron barely interacts with the outer shells so that the final state wave functions, in the region of the $2p$ subshell, are very close, as well. In the atomic case, the $2p$ cross section, 20 eV above threshold, is about 9 Mb [33], as compared to roughly half that value for the photodetachment experiment [28], as seen in Figure 1. This suggests that the absolute normalization of the experimental photodetachment cross section might be too small by a factor of two or so. Furthermore, multiplying the experimental cross section by a factor of two would bring that scaled cross section into quite good quantitative agreement with the R-matrix calculation.

To further investigate this point, a calculation of $2p$ ionization in neutral Na has been performed using the same methodology as the Na$^-$ calculation, and the results are shown in Figure 2 as compared

to the Na⁻ results. It is clear that the Na 2*p* cross sections at higher energies agree quite well with the Na⁻ results, as suggested above, in both length and velocity formulations. Furthermore, the calculated Na cross sections are in substantial agreement with earlier experimental and calculated results for neutral Na [33]. For these reasons, then, it would seem to be a really good idea to look again experimentally at the Na⁻ photodetachment cross section, particularly at the absolute normalization.

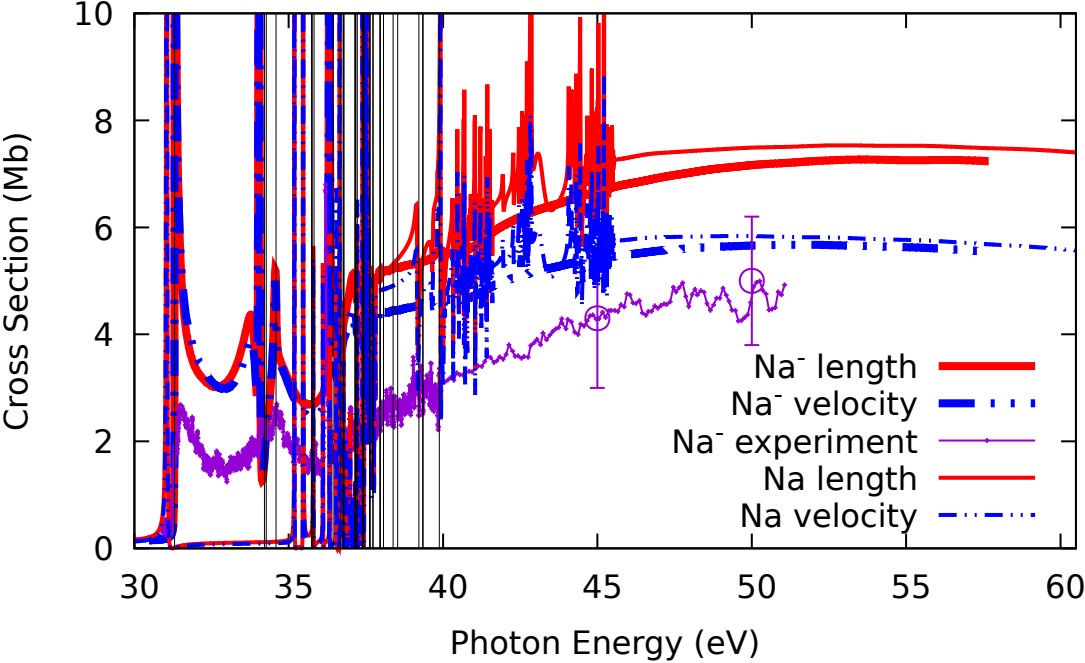

**Figure 2.** Present R-matrix Na photoionization cross sections compared to present R-matrix Na⁻ photodetachment cross sections, in length and velocity gauges. The calculated inner-shell thresholds are shown as vertical black lines.

The resonance just above 36 eV is difficult to see in Figure 1, so a blow-up of that region of the cross section in given in Figure 3. This resonance lies just below the $2p^53s4s$ doublet thresholds, and we conclude that it is a Feshbach $2p^53s4s^2$ $^1P$ resonance. The reason that $2p \to 4s$ photoexcitation is chosen, rather than the usually dominant $2p \to 3d$ "giant resonance" transition, is that the 3*d* orbital in the Na⁻ $3p^63s3d$ and $2p^53s^23d$ state is so much more diffuse than the tightly-bound 2*p* orbital that the *dynamic* transition factor is diminished in the latter transition, despite the larger $2p \to 3d$ *geometric* factor. Considering the one-electron frozen-core transition, this resonance must be associated with the $(2p^53s)[^1P]4s(^2P)$ threshold, as opposed to the $(2p^53s)[^3P]4s(^2P)$ threshold, as suggested earlier [19]. The theoretical resonance is seen to lie about 0.1 eV above the experimental resonance, and the shapes are more or less the same. However, it is also seen that the theoretical resonance peak is about a factor of two larger than the experimental peak, further suggesting that the absolute normalization of the experimental cross section might be off by as much as a factor of two. Since this difference is roughly the same over the entire energy range, even in the threshold region, it was thought that the PCI effect ia not great in this case, so it has been omitted.

A comparison with the two previous calculations [20,21] is given in Figure 4, along with the experimental results. As mentioned earlier, neither of the earlier calculations showed a sharp Feshbach resonance just above 36 eV because neither of them included the relevant detachment-plus-excitation channels. Both show resonant-like behavior near threshold that is not associated with Feshbach resonances. It is also of interest to note that both earlier calculations predict cross sections that are closer to the magnitude of the experiment, which argues somewhat against our notion that the absolute magnitude of the published experimental results is off by a factor of two. Again, an experimental re-examination of the Na⁻ photodetachment cross section is required to settle the issue.

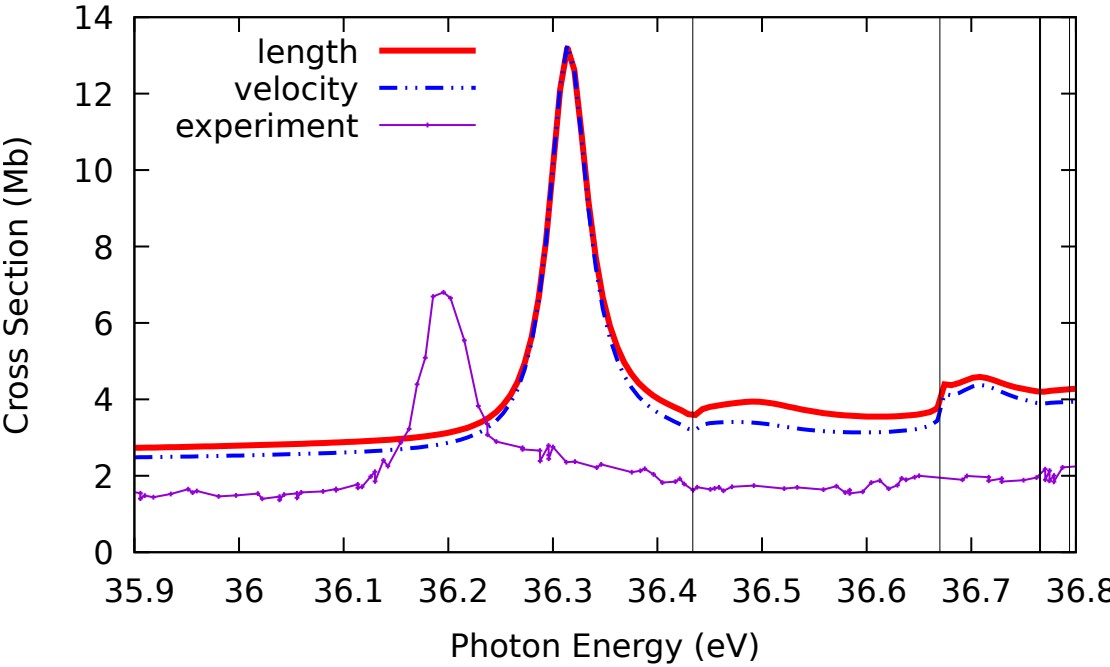

**Figure 3.** Photodetachment of Na$^-$ near the Feshbach resonance just above 36 eV. The calculated inner-shell thresholds are shown as vertical black lines.

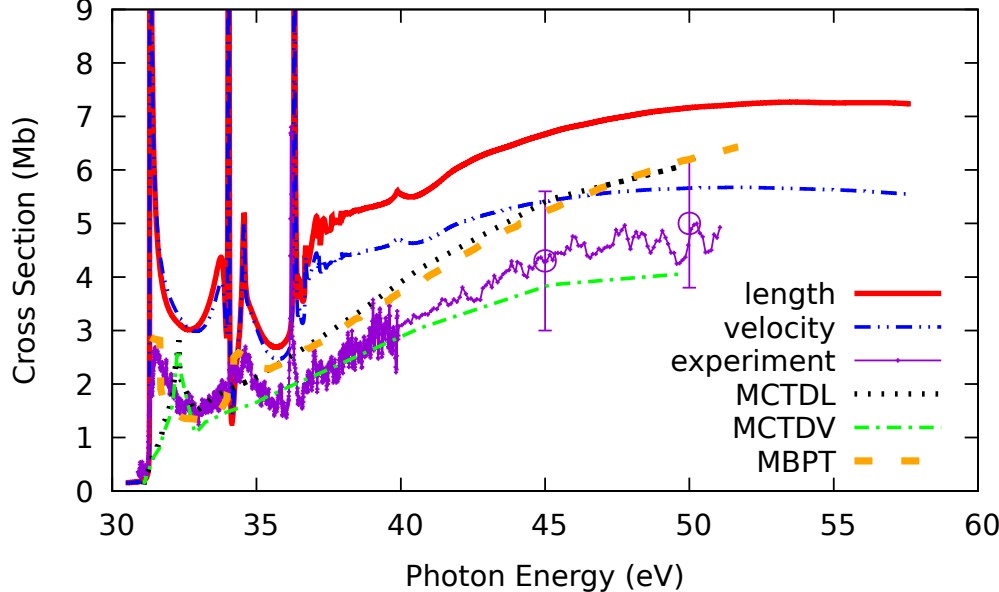

**Figure 4.** Photodetachment of Na$^-$: Present R-matrix cross sections compared to earlier theoretical results: the Multiconfiguration Tamm-Damcoff results [21] in length (MCTDL) and velocity (MCTDV) forms, as well as many-body perturbation theory (MBPT) results [20].

Another point of importance is the agreement between length and velocity in the present calculation. From the inner-shell detachment threshold to about 37 eV, the agreement between the two gauges is seen to be excellent. This agreement strongly suggests (but does not prove) that the present calculation is reliable in this energy region. At the higher energies, the agreement is not quite as good, with a roughly 20% difference between length and velocity gauge results, but is sill not terrible. The disagreement in this region is likely due to the omission of the myriad of $2p^5nln'l'$ states of Na in the close-coupling expansion.

## 4. Summary and Conclusions

The inner-shell photodetachment cross section of Na$^-$ has been calculated using R-matrix methodology. The results show excellent *qualitative* agreement with experiment, including the sharp Feshbach resonance just above 37 eV that was absent in previous calculations. The disagreement with experiment at the first 2$p$ threshold is mainly due to the PCI effect which is not included in the present calculation. Overall, the calculated cross section is about a factor of two larger than the experimental result, and this is problematical. Owing to the agreement of length and velocity gauges in the calculation, along with the agreement of the theoretical Na$^-$ cross section with the 2$p$ photoionization cross section of neutral Na, it is suggested that the measured inner-shell photodetachment cross section of Na$^-$ is too small by a factor of about two; it is strongly suggested that this matter be re-examined experimentally.

Many-body correlation effects were seen to be not merely important but crucial to the behavior of the inner-shell Na$^-$ photodetachment cross section. The initial state of Na$^-$ is not even bound without the inclusion of correlation. And the various structures seen in the photodetachment cross section are intimately tied up with correlation in the final-state wave function, effects that are absent (or much diminished) in the corresponding photoionization of neutral Na. The situation can be thought of in the framework of a simple conceptual model. An ionizing photon excites inner-shell electron, and the photoelectron emerges through the cloud of outer-shell electrons. On its way out, the photoelectron can scatter inelastically off an outer-shell electron, giving up some of its energy and exciting the outer-shell electron, resulting in a final state of photoemission-plus-excitation; from a quantum mechanical point of view, this is just interchannel coupling. For the photodetachment process, this inelastic excitation process is very strong, since the outer-shell electrons are bound so weakly in a negative ion, yielding a cross section replete with significant inner-shell photodetachment-plus-excitation channels and structures. For the neutral photoionization process, the same model applies, but the outer-shell electrons are much more tightly bound so that the probability of the excitation process is very much smaller. For this reason, then, although the photoionization-plus-excitation process does, in fact, occur for neutral Na, it is so much smaller than the main photoionization cross section that it is hardly noticeable. This shows clearly why the study of photodetachment is such an excellent venue for investigating many-body correlation in atomic systems.

**Author Contributions:** Conceptualization, T.W.G., H.-L.Z., A.H. and S.T.M.; methodology, T.W.G., H.-L.Z., A.H., M.F.H. and S.T.M.; software, T.W.G., H.-L.Z., A.H.; validation, T.W.G., H.-L.Z., A.H., M.F.H. and S.T.M.; formal analysis, T.W.G., H.-L.Z., A.H., M.F.H. and S.T.M.; investigation, T.W.G. and S.T.M.; resources, T.W.G. and S.T.M.; data curation, T.W.G., H.-L.Z., A.H. and M.F.H.; writing—original draft preparation, T.W.G. and S.T.M.; writing—review and editing, T.W.G., H.-L.Z., A.H. and S.T.M.; visualization, A.H.; supervision, T.W.G. and S.T.M.; project administration, T.W.G. and S.T.M.; funding acquisition, T.W.G. and S.T.M. All authors have read and agreed to the published version of the manuscript.

**Funding:** T.W.G. was supported in part by NASA (NIX11AF32G). S.T.M. was supported by the U.S. Department of Energy, Office of Basic Sciences, Division of Chemical Science, Geosciences and Biosciences under Grant No. DE-FG02-03ER15428.

**Conflicts of Interest:** The authors declare no conflict of interest.

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
