# Peer review of "Inner-Shell Photodetachment of Na− Using R-Matrix Methods"

_atoms, doi:10.3390/atoms8030060_

Round 1
Reviewer 1 Report
The paper presents a calculation on the near-threshold inner-shell photodetachment cross sections on the negative sodium ion. This calculation is based on the Belfast R-matrix methodology. Atomic orbitals are generated using the atomic structure program CIV3. The presented results show the qualitative agreement with the experiment. The paper is well-written, the topic is a research subject of interest and the authors employ a simulation approach, in which they can be considered as experts according to the development in their latest article [Ref. 30]. I think the paper can be accepted for publication in Atoms after minor modification based on the comments stated below:
- Why does the present calculation neglect the post-collision interaction (PCI) effect?
- How the present study can be improved for quantitative agreement with the experiment?
Author Response
Reply to the Referees
We thank the referees for their piositive comments and sensible suggestions. Our specfic resonses are detailed below:
Referee 1:
1. Why does the present calculation neglect the post-collision interaction (PCI) effect?
Reply: This is a cogent question, and it should have been commented upon in the manuscript. Since the experiment and the the calculation were so far off, by a factor of 2 or so, including the threshold region where PCI should be important, it seemed that the PCI effect was not very important in this case, so it was omitted. This has been added to the manuscript with sentences at the end of the second and sixth paragraphs of the Results and Discussion section. We thank the referee for pointing out this omission.
2. How the present study can be improved for quantitative agreement with the experiment?
Reply: We suggest in the manuscript that for the principal disagreement with experiment, the overall normalization, that it is the experiment that is incorrect, and this inference is supported by comparison with neutral Na, as detailed in the manuscript. Other than this, the primary features of the experiment are reproduced pretty well, although, as suggestesd in the manuscript, some of the features predicted by the calculation are obliterated by the experimental bandwidth. Thus, a call for additional experimental work with finer resolution is also made in the manuscript. In other words, we believe the ball is in the experimentalist's court.
Referee 2:
1. Although the discussion of the removal of pseudoresonances has
been described previously in Ref. 24, for the sake of completeness it would be useful to rewrite and extend the single long sentence on p. 2.
Reply: The discussion of pseudoresonance removal has been re-worked and extended, as suggested.
2. On p. 7, the sentence "An ionizing photon is absorbed by an
inner-shell electron ..." should be reworded since electrons
cannot absorb photons. Perhaps say instead "An ionizing photon
excites an inner-shell electron ... ".
Reply: The wording has been changed as suggested.
Reviewer 2 Report
This is an interesting and well-written article presenting
R-matrix calculations for the inner-shell photodetachment of
sodium. The paper highlights the important role played by
electron correlation in the photodetachment cross section. The
overall qualitative agreement with experiment is excellent, and
the paper identifies the origin of many of the resonances. The
main disagreement is in the overall absolute values of the cross
sections. Agreement between the length and velocity forms
suggests that the experimental normalization may be incorrect.
The paper should certainly be accepted for publication with the
following minor changes:
1. Although the discussion of the removal of pseudoresonances has
been described previously in Ref. 24, for the sake of completeness it
would be useful to rewrite and extend the single long sentence on
p. 2.
2. On p. 7, the sentence "An ionizing photon is absorbed by an
inner-shell electron ..." should be reworded since electrons
cannot absorb photons. Perhaps say instead "An ionizing photon
excites an inner-shell electron ... ".
Author Response

(The authors gave the same response as above.)
